# Evidence-based practice by physiotherapists in UAE: Investigating behavior, attitudes, awareness, knowledge and barriers

**Hamda AlKetbi**[1], **Fatma Hegazy**[2,3], **Arwa Alnaqbi**[4], **Tamer Shousha**[2,5]*

**1** Physical Medicine & Rehabilitation Department, Rashid Hospital, DHA, Dubai, United Arab Emirates, **2** Department of Physiotherapy, College of Health Sciences, University of Sharjah, Sharjah, United Arab Emirates, **3** Department of Physical Therapy for Growth and Development Disorders in Children and Its Surgery, Faculty of Physical Therapy, Cairo University, Cairo, Egypt, **4** Physiotherapy Department, Ministry of Health and Prevention, Dubai, United Arab Emirates, **5** Department of Physical Therapy for Musculoskeletal Disorders and Its Surgery, Faculty of Physical Therapy, Cairo University, Cairo, Egypt

☯ These authors contributed equally to this work.

* tshousha@sharjah.ac.ae

**Data Availability Statement:** All relevant data are within the paper and its Supporting Information files.

## Abstract

Evidence-based practice (EBP) is an important factor determining the quality of healthcare. The field of physiotherapy is still limited by indirect access in several countries including the United Arab Emirates (UAE) which creates added pressure to justify the merit in its practitioner's capabilities. This study explores the behavior, attitudes, awareness and knowledge towards EBP among practicing physiotherapists in the UAE. It also enquires about their perception of the barriers in the implementation of EBP. Using a questionnaire survey of 258 physiotherapists, results show that the awareness of EBP is limited to a few terms including EBP, systematic literature review, and random trials while other terms associated with scientific studies are not known well. The attitude towards EBP was found to be significantly related to the knowledge of EBP (r = 0.208) and the perception of barriers to it (r = 0.156). The EBP behavior was found positively related to its knowledge (r = 0.134) and the perception of barriers (r = 0.216). The physiotherapists prefer to use their own experience and books and research articles to apply EBP but do not consider their peers to be as worthy sources as the others. However, their attitudes towards EBP are largely positive though their perception of barriers grows with better knowledge and understanding of EBP. The barriers in the implementation of EBP are a lack of research knowledge and skills, time, support, and resources which indicate opportunities for the decision-makers to improve the adoption of EBP among these professionals. This study concluded that although physiotherapists in the UAE claim awareness about EBP, their knowledge is limited to a few key terms whereas, attention is needed to improve EBP knowledge and practice.

## Introduction

Physiotherapy is as much of an art as it is a science. Evidence-based practice (EBP) can help in achieving both aspects of this discipline as the scientific evidence from scholarly studies and the primary data collected from the patients and their caregivers can assist physiotherapists in

**Funding:** The authors received no specific funding for this work.

**Competing interests:** The authors have declared that no competing interests exist.

their diagnosis, as well as, the creation of specific treatment plans. The practice of physiotherapy improves through EBP as it has been linked to better health outcomes in the patients as their needs are identified and incorporated in the tailored treatment plans [1, 2]. Moreover, physiotherapists gain valuable experience by involving themselves in EBP [3]. Further, the growing awareness among service users is also bringing a much-needed shift in the provision of healthcare as they demand better quality of service which necessitates better collaboration and coordination among different stakeholders in the system [2, 4]. In addition, EBP is finding more clinical evidence as studies indicate that physiotherapists who are employing EBP are bale to perceive better quality of healthcare provided by them [5, 6]. Therefore, the evidence for the significance of EBP for physiotherapy is strong and growing.

On the other hand, there are several gaps in the application of EBP by physiotherapists due to persisting challenges. Frantz and Rowe [7] have pointed out that the skills of the physiotherapists in appraising research literature and using it in their practice has a direct impact on its success. Administrators and hospital managers have been reported to be lacking in their support for the availability of resources, time, and encouragement for EBP [8, 9]. Physicians including physiotherapists remain so busy in their daily routines that using EBP is kept on a backburner [10]. Resources in terms of availability of research journals or evidence summaries is also found wanting [11]. Moreover, studies indicate that the positive attitude of physiotherapists towards EBP itself is missing [12].

In this context, it becomes important to assess the awareness, attitudes, and behaviors of the physiotherapists from the UAE regarding EBP. The researcher has chosen the UAE for the setting of this study as despite being one of the more developed nations with a high per capita income, its healthcare is perceived to be weak in its implementation of EBP [9]. Moreover, the practice of physiotherapy in the country is unique from other developed and developing nations as it does not offer direct access to its physiotherapists in the public healthcare while the discipline itself is marked by low organization among practitioners who themselves are highly diverse, belong to over fifty nationalities [13].

EBP is defined as, "the conscientious, explicit and judicious use of current best evidence in making decisions about the care of individual patients" [8]. It originated with the assertion of Mary Richmond who reported that using research evidence to make decisions in healthcare can help to reinforce best practices and bring about reforms in the practice of physiotherapy [14]. She was the first one to point to that science and art have to move together for better practice of this discipline [15]. EBP has been visualized a process which begins with the articulation of a need or problem, moving on to a search for possible options that can satisfy the need, evaluating them to see if they can meet the need, implementing the best solution, and reviewing it to see if the expected benefits from its application have been achieved [16, 17]. Fig 1 shows this process in detail.

It is believed that this process can be of immense use in medicine [18], with researchers even asserting that it is not correct to forego EBP at all [19]. Even the field of physiotherapy is reported to benefit from it as within a set of nine competencies suggested by the Europe Region World Physiotherapy [20], the ability to research available evidence and analyze it in the right context, thereby, applying EBP in their work is also included.

Despite this significance of the concept, EBP practice remains low among physiotherapists due to lack of resources, time, lack of training among practitioners, and low availability of sources of evidence [12, 21–23]. It is reported that physiotherapists grow to feel more underconfident about their ability to apply EBP and have less positive feelings about it after gaining more work experience [3]. Physiotherapists have been reported to even state that they do not have any positive feelings towards EBP as they feel it will hamper their freedom to practice using their own judgment [24]. Such studies show that there may be deep-rooted attitudinal

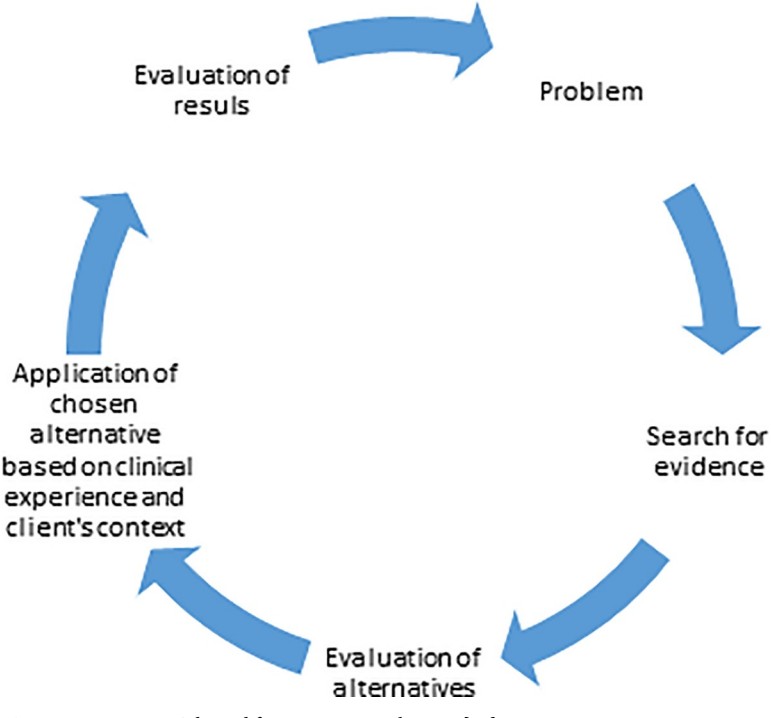

**Fig 1. EBP process.** Adopted from Haynes et al., 1997 [17].

barriers against EBP among the physiotherapists which may prevent them from applying EBP even if they are aware of it.

Studies have even indicated that among all kinds of healthcare professionals, it is the physiotherapists who lag behind others in applying EBP [25, 26]. Studies report that physiotherapists lack support and opportunities to discuss literary evidence with their peers which affects their implementation of EBP [27].

Other barriers include a lacking organizational culture which encourages and promotes the use of EBP. Quinn et al. [28] reported that facilitator stewardship, evaluation strategies, supportive culture, organization receptivity, and facilitated change management are needed for the appropriate implementation of EBP. A lack of awareness of EBP is one of the foremost barriers to applying it. In a study located in Sweden, only 12 to 36% out of a sample size of 833 were reported to be aware about EBP [29]. Such low awareness of the concept is bound to affect its uptake among the physiotherapists. This prompted the researchers to explore and substantiate the possibility of leadership support to assist in popularizing EBP [30]. More such studies are needed to explore such interventions in varied contexts.

It is for this purpose that this research has explored the perception of the physiotherapists practicing in the UAE about EBP. The following research objectives were, therefore, proposed:

1. To identify the current awareness and knowledge towards EBP among physiotherapists practicing in the UAE.

2. To assess the attitude and behavior towards EBP among physiotherapists practicing in the UAE.

3. To identify the barriers towards effective implementation of EBP in the UAE and the strategies that can be adopted to resolve them.

## Methods

This study was approved by the University of Sharjah Research Ethics Committee approval no.: REC-20-06-14-01-S. A descriptive cross-sectional design with a sample of convenience was adopted. The participants of the study were all physiotherapists working in the UAE either in clinical practice or an academic setting. Both genders were considered in this study. Physiotherapy Interns and non-graduating physiotherapy students were excluded. All subjects signed an informed consent attached with the questionnaire prior to starting the study.

### UAE physiotherapists

The level of organization of the UAE physiotherapists is low as only 314 members are listed with the UAE wing of World Confederation of Physiotherapy (WCPT) though they estimate the total numbers of practicing physiotherapists to be nearly 3000 [31]. The UAE healthcare system has suffered due to its weaknesses in the past when projections reported that $17 billion were spent by its own citizens who preferred to get treated abroad [32]. As a result, the policy-makers are focused on converting the country into a medical tourism hub and also improve the delivery of healthcare services through apps like the Dubai Health Experience (DXH) which allows Gulf cooperation council (GCC) citizens to access healthcare smartly [33]. This strategy will allow the country to retain the revenue spent abroad by its citizens and also of other GCC countries and it is already bearing fruit as 26% of this money is now spent within the UAE [34]. There is a special focus on physiotherapy as well in the 2030 Dubai Industrial Strategy with plans to set up 34 factories that will build equipment to aid the discipline [35]. However, this scaling of infrastructure and resources is not complete till the government also pays attention to critical internal factors like the improvement of EBP application and also direct access to the profession. Furthermore, the level of educational facilities, the curriculum, and the quality of teaching are also instrumental in instilling the knowledge and the belief in EBP. This need for an aligned educational framework has also been reported to be weak in certain countries [36]. As a result, studies are needed to explore which particular barriers affect the practice of EBP in physiotherapy so that appropriate steps can be taken to correct them.

### Design

As indicated in the literature review, studies about the application of EBP among the physiotherapists have not yet been conducted within the UAE, though one study explored its awareness about student physiotherapists [9]. As a result, this study has adopted a cross-sectional research design to explore the awareness, attitudes, and behaviors of the practicing physiotherapists so that decision-makers can understand how best to improve its implementation. To achieve this research aim, a questionnaire survey was designed using the one employed by Alshehri et al. (2017). This questionnaire was further refined by adding a question regarding the years of experience [1] and validated through a pilot study with 10 physiotherapists. The population of this study consisted of all practicing physiotherapist of the UAE with at least a year's experience in the profession and an ability to understand and answer the questionnaire in English.

### Procedure

The data collection happened over the course of four weeks beginning in March 2020 when the questionnaires were emailed to the selected participants in the sample with a cover letter explaining its purpose and seeking their permission to join the study. The physiotherapists were identified from the major healthcare institutions and clinics located within the UAE

through physical visits and their websites which helped in creating a sampling frame and employing random assignment to identify the first 300 participants for the survey. Three hundred questionnaires were sent out and 258 were finally received with all complete entries.

The collected data was collated in an Excel file which was later subjected to data analysis using IBM's SPSS software (IBM SPSS Statistics for Windows, Version 24.0. Armonk, NY: IBM Corp.).

The reliability and validity of the questionnaire were assessed using the Cronbach alpha and the Spearmans' correlation coefficient respectively with a P value set at P<0.01. The results of the demographic variables' influence on the awareness, knowledge, attitude, and behavior towards EBP were analyzed using MANOVA where the means and standard deviations of the answered questions of the subscales were calculated and finally the relationship between different variables was assessed using the Pearsons' correlation coefficient.

The survey consisted of 15 closed-ended questions in the following sequence (S1 Appendix):

1. Demographic data: (age, gender, education, university offering the degree, job title and work placement and years of experience)

2. Behavior: A Likert scale ranging from always (1) to never (5) used to answer six questions. These questions determine sources used daily by physiotherapists within their clinical practice.

3. Attitudes: A Likert scale ranging from strongly disagree (1) to strongly agree (5) to answer four questions were used to determine the attitudes towards the use of research in practice.

4. Awareness: To assess the physiotherapists' awareness of EBP, 14 questions related to research terminologies were used. Five answer options for each question are available, where each answer had a different score: never heard of it (score = 0), have heard of it but do not understand it (score = 1), understand it a little (score = 2), understand it very well (score = 3), understand it completely and could explain it to others (score = 4). Higher scores indicated a higher level of awareness towards EBP implementation.

5. Knowledge: Six questions were set to examine physiotherapists' knowledge in relation to EBP implementation. Each had three answer options: (1) agree, (2) disagree and (3) unsure. Only the second answer (disagree) was considered valid for all six items; receiving two points with a maximum score of 12. Greater scores indicated a greater level of EBP knowledge.

6. Formal training: To identify if physiotherapists previously received formal EBP training from as part of university education or courses of continurd education organized by UAE healthcare providers or offered online by accredited bodies.

7. Barriers: Six items about possible barriers to EBP implementation assessed by a ranking scale ranging from 1 (the least important barrier) to 10 (the most important barrier).

## Results

The reliability of the questionnaire was then assessed using the Cronbach's alpha. Table 1 shows the individual scale's Cronbach alpha values, as well as, the value for the entire scale.

The construct validity was assessed using Spearman rank coefficients between the modified questionnaire and the original used by AlShehri et al., 2017. The values were interpreted as follows: excellent relationship, >0.91; good, 0.90–0.71; moderate, 0.70–0.51; fair, 0.50–0.31; and little or none, < 0.30 The level of significance in all tests was set at p < 0.01.

**Table 1. Reliability analysis.**

| Variables | Item Numbers | Cronbach's Alpha |
|---|---|---|
| Attitude towards EBP | 4 | .936 |
| Behaviour towards EBP | 6 | .661 |
| Awareness of EBP Terms | 14 | .946 |
| Knowledge of EBP | 6 | 0.72 |
| Barriers to EBP | 6 | .884 |
| Total | | **.867** |

The only scale as shown in (Table 1) which has fetched a Cronbach's alpha value of less than 0.7 is the usage of EBP but it is disregarded as all four items of the scale are critical to understanding it and further analysis revealed that removing any of them was not making a substantial effect on the scale's reliability. At the same time, the entire questionnaire's internal reliability is high as denoted by the Cronbach's alpha value of 0.867. Therefore, the questionnaire satisfied the reliability criteria.

The validity of the questionnaire was reported as (0.92) revealing an excellent positive correlation with the original questionnaire.

The demographic data collected in the first part of the questionnaires showed that more women (58.9%) than men (41.1%) answered the questionnaire (Table 2). More respondents belonged to the 31–35 years (33.7%) and 36–40 years' (31%) age groups. The wide variety of nationalities seen in the physiotherapy discipline was also seen in the respondents with 26

**Table 2. Detailed analysis of questionnaire items.**

| Factor | Item | Statement | Mean | Standard Deviation |
|---|---|---|---|---|
| **Awareness of EBP Terms** | 1 | EBP as a term | 4.16 | 1.042 |
| | 2 | EBP cycle/steps | 3.77 | 1.238 |
| | 3 | Quality of evidence | 3.91 | 1.095 |
| | 4 | Systematic review | 4.02 | 0.954 |
| | 5 | Randomized controlled trial | 4.00 | .968 |
| | 6 | PICO | 3.15 | 1.328 |
| | 7 | Critical Appraisal | 3.70 | 1.14 |
| | 8 | Forest plot | 2.67 | 1.3 |
| | 9 | Relative risk | 3.19 | 1.26 |
| | 10 | Likelihood ratio | 3.09 | 1.32 |
| | 11 | Confidence interval | 3.4 | 1.24 |
| | 12 | Effect size | 3.34 | 1.32 |
| | 13 | Risk of bias | 3.57 | 1.19 |
| | 14 | Healthcare databases such as MEDLINE, PEDro, etc. | 3.75 | 1.12 |
| **Attitude of EBP** | 1 | Understanding of research methods and research designs is important in physiotherapy practice. | 4.24 | 1.07 |
| | 2 | Research theory and methodology should be included in the physiotherapy curriculum. | 4.19 | 1.15 |
| | 3 | Physiotherapists need to read relevant articles regularly to update their knowledge. | 4.48 | 1.07 |
| | 4 | Physiotherapists should apply treatments that are supported by evidence. | 4.45 | 1.01 |
| | | | Mean | Standard Deviation |
| **Behaviour towards EBP** | 1 | My personal experience | 4.39 | .798 |
| | 2 | My colleagues' opinions | 3.79 | .993 |
| | 3 | My supervisor's or expert opinions | 4.09 | 0.912 |
| | 4 | Internet | 3.95 | 1.008 |
| | 5 | Books | 4.28 | 0.856 |

nationalities participating in the survey though the highest numbers belonged to the South East Asia (43.4%), the UAE (26%), and the other middle eastern countries (17.9%). More than half of the sample had completed a Baccalaureate degree (58.1%) followed by Master's (38.8%). More respondents were affiliated with the Ministries (37.2%), than the Dubai Health Authority (24.4%), the private health sector (16.3%), the Abu Dhabi Health Authority SEHU (15.5%) and academics (3.9%). More than half of the sample were Senior physiotherapists (51.9%) with the juniors also represented well (34.5%). The greatest number of respondents had 6–10 years of experience (31.8%) followed by 11–15 years (29.5%), and 1–5 years (19.7%). As a result, the sample represented diverse demographic characteristics of the population.

The first subscale enquired about the awareness of EBP among the physiotherapists using a list of fourteen terms associated with research articles and experimental studies. Among these terms EBP itself scored the highest (Mean = 4.16, SD = 1.04), followed by systematic review (Mean = 4.02, SD = 0.954), and randomized controlled trial (Mean = 4.00, SD = 0.968). The terms which fetched the lowest ratings were forest plot (Mean = 2.67, SD = 1.3), Likelihood ratio (Mean = 3.09, SD = 1.32), and PICO (Mean = 3.15, SD = 1.33).

The results of the questionnaire items show that the respondents possessed a positive attitude largely as indicated by the above 4 Mean values shared for each of the items in the attitude subscale. The respondents particularly were more favorable towards the need to continuously refresh their knowledge by remaining up to date with research articles (Mean 4.48, SD = 1.07) though the belief that physiotherapists should apply EBP supported treatment was also highly rated (Mean = 4.45, SD = 1.01). Previous studies have also reported that physiotherapists believe EBP to be an important aspect of their profession [7, 11].

The knowledge subscale was designed to elicit the understanding of the respondents about EBP. Among the six items in the subscale, "EBP is a process of systematic investigation to generate knowledge and test theories" found the most numbers of respondents agreeing with it (84.9%). The next highest level of agreement was reported for "The main aim of EBP is to identify the causes of research problems and how to solve them" (76.7%). The respondents rated "Patient's values and preferences are not one of the main requirements of EBP" (67.8%) and "Physiotherapy interventions are mostly supported by EBP" (68.2%) relatively lower.

The next subscale in the questionnaire was designed to understand who the awareness and knowledge of EBP was translating into behavior. The Mean values were found to be lower than those reported for earlier subscales. The respondents indicated that they preferred to apply own experience (Mean = 4.39, SD = 0.798), books (Mean = 4.28, SD = 0.856) and research articles (Mean = 4.2, SD = 0.796) over their peers' opinion which received the lowest scores (Mean = 3.79, SD = 0.993). As working in collaboration with colleagues has been reported to provide valuable evidence for EBP practice, this result is an important finding [3, 37, 38].

The questionnaire also enquired about the barriers to the application of EBP using a list of six barriers which could be ranked from 1 to 10 with 10 representing the most influential barrier. Lack of research knowledge and skills (Mean 7.15, SD = 2.6), lack of support and encouragement (Mean = 7.08, SD = 2.57), and lack of funding and resources such as access to databases and journals (Mean 7.02, SD = 2.67) were the top rated barriers affecting the implementation of EBP. Lack of interest (Mean 5.98, SD = 3.07), however, was not believed to be as important.

Further, the variables of awareness, knowledge, attitude, behavior, and barriers to EBP were correlated with each. The results of Pearson's correlation between these variables are shown in the Table 3.

As shown in the table above, the awareness of EBP is not related to the attitude towards it, its knowledge, or the perception of barriers. However, it is positively related to positive EBP

**Table 3. Pearson's correlation between study variables.**

| Variables | Awareness of EBP | Attitude towards EBP | behaviour towards EBP | Knowledge of EBP | Barriers to EBP |
|---|---|---|---|---|---|
| Awareness of EBP | | .056 | .128* | .016 | .093 |
| Attitude towards EBP | | | -.024 | .208** | .156* |
| behaviour towards EBP | | | | .134* | .216** |
| Knowledge of EBP | | | | | .093 |

* Significant at 0.05 level.

**Significant at 0.01 level.

behavior though the effect size is low (r = 0.128*) [39]. The attitude towards EBP is significantly related to the knowledge of EBP (r = 0.208**) and the perception of barriers to it (r = 0.156*) though both relationships are also low in effect size. The EBP behavior is positively related to its knowledge (r = 0.134*) and the perception of barriers (r = 0.216**) though again with low effect size. These results show that with the practice of their profession, the physiotherapists' perception of barriers to EBP increases.

Regarding education, the qualifications of the respondents were significantly related to the dependent variables though only knowledge of EBP differed significantly among the dependent variables. The effect size for this relationship was again low which explained 10.7% of variance in scores. The PhD holding physiotherapists had a better knowledge of EBP (Mean 66.5) than the others with each level of education contributing to higher knowledge among the physiotherapists. For instance, the Baccalaureate students revealed a Mean value of 46.49, Master's 50.75, and DPT 53.88. This is an encouraging trend as it shows that the educational qualifications are contributing to better knowledge of EBP among the physiotherapists.

In addition, results revealed that gender did not influence the opinions of the physiotherapists towards EBP for their awareness, knowledge, attitudes or behavior. However, gender was found to have an impact on the perception of barriers in the path of implementation of EBP. Though this relationship was found to have a small effect size, the women (42.19±12.41) perceived more barriers for EBP compared to men (38.2±13.04).

Similarly, the perception of Barriers to EBP were significantly related to the age of the physiotherapists. Further exploration revealed that the differences lay between the youngest group of 20–25 years old physiotherapists and the 31–35 years and 36–40 years old physiotherapists. The effect size was found to be small with the 20–25 years old physiotherapists (30±14.39) having significantly lower perception of barriers compared to the 31–35 (41.66±13.19) and 36–40 years old physiotherapists (41.90±12.31). This result clearly shows that with younger age, the physiotherapists are encountering more barriers in the implementation of EBP.

## Discussion

This study has shown that the awareness of EBP among the physiotherapists is limited to a few terms while their knowledge of scientific studies is not deep. With earlier evidence also indicating a similar lacuna, it is important that decision-makers take active steps to introduce EBP in educational and training curricula [8, 12, 21, 22].

Among the fourteen terms associated with EBP, the participants of this study rated only three above a rating of 4 which signified that they understood the term very well. These three terms include EBP as a term (4.16±1.04), systematic review (4.02±0.954), and randomized controlled trial (4.00±0.968). Ten other terms fetched Mean values above 3 which signified that they understood the terms a little. The lowest rated terms were forest plot (2.67±1.3), Likelihood ratio (3.09±1.32), and PICO (3.15±1.33).

The awareness of EBP in this study was seen superficial and is not yet supported through a deep review of studies in the field of physiotherapy and medicine. The lower ratings to terms like confidence intervals, effect size, and risk of bias indicated that the physiotherapists are not yet proficient to assess the rigor of quantitative research designs.

The above results came into agreement with earlier studies describing awareness of EBP among physiotherapists as below expectations [12, 21, 22]. Similarly, an enquiry of EBP practice among various healthcare professionals in the UK revealed that their understanding of the term and actual usage of EBP were infrequent [21]. These healthcare professionals chosen were new to the profession with only two years' experience of working as physiotherapists showing that they had failed to learn about EBP during their education or the practice of their profession.

A better proportion was reported by Ramírez-Vélez et al. (2015) who reported that in their sample of 1064 Colombian physiotherapists, 78.1% knew about EBP and its guidelines. Among these, 71% strongly agreed that EBP is necessary for physiotherapists. Scurlock-Evans et al. (2014) also reported that the combined indications from thirty-two studies were that the physiotherapists, despite positive attitudes of EBP, were not applying it well.

It is important to reiterate here that unlike this study, the present research involves a majority of physiotherapists with only a Bachelor's degree (N = 150). In a study conducted on similar lines to the present one, Alshehri et al. (2017) who surveyed 604 Saudi Arabian physiotherapists, reported that only 29.8% of the participants completely understood the term EBP. Moreover, 23.1% had never heard of the term before showing that one-quarters of the population was completely unaware of EBP. In this study, 10 participants have chosen the option one which signified that they had never heard about EBP before translating into a percentage of 3.88. Therefore, our results were much better than those reported by Alshehri et al. (2017). However, another 15 participants (5.81%) had heard about the term but did not understand it well. Therefore, a gap can be seen in the awareness of the physiotherapists with regard to EBP as a term which becomes even more prominent when we explore their understanding of allied terms. Like Alshehri et al. (2017), in this study forest plot remained the least understood term.

In addition, the training received by the participants for EBP indicated their knowledge and awareness of the term. Only 34.5% of the sample had received such a training. Amongst this group, 25.2% had attended a EBP course as part of their University education which included more than 20 hours of instruction. The rest of the group had only attended a comprehensive course of 11 to 20 hours' duration (2.7%), a short course of 3 to 10 hours' duration (3.5%), or only a lecture of 1 to 2 hours (3.1%).

As training in EBP has been linked to higher adoption and implementation rates, this level of training is one of the areas needing further attention by the decision makers [38].

Another important criteria for successful employment of EBP is a positive attitude towards it [7, 11] which leads to self-reported EBP use [40]. Contrarily, negative attitudes can make the physiotherapists feel trapped to use EBP and consider a burden [41, 42] making it a barrier [12, 43]. This study has shown that physiotherapists in the UAE hold positive attitudes towards EBP which can be leveraged by the decision-makers to encourage them to adopt it more frequently in their practice.

Further as identified in this study, lack of resources, lack of time, lack of library resources, knowledge, and skills, lack of support through designated personnel and organizational climate, have been identified by many earlier studies [22, 43, 44]. As a result, urgent steps are needed to address them at the macro level.

Finally, the results of this study indicate an interaction between the study variables which are visualized in the following figure (Fig 2).

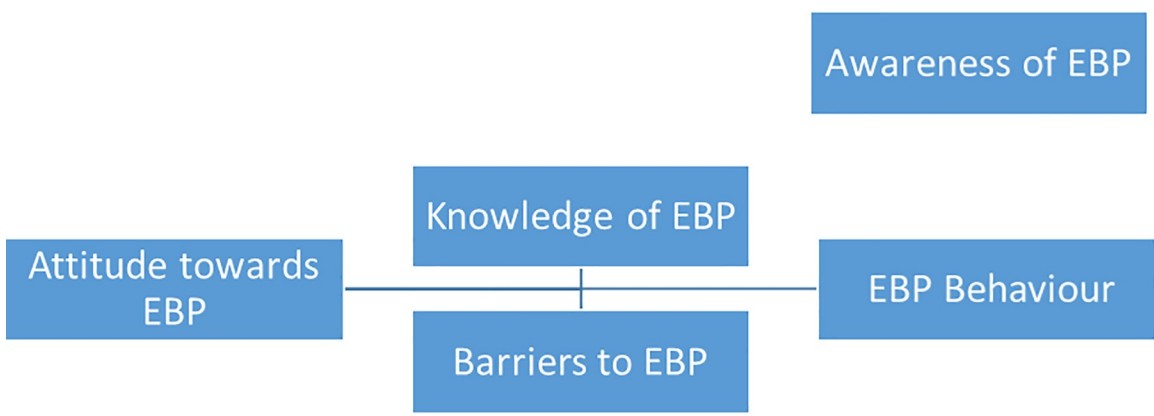

**Fig 2. Interaction between EBP domains.**

The awareness of EBP leads to EBP behavior while positive attitudes improve its knowledge and help get a more realistic idea of the barriers of EBP. All of these factors lead to more positive behavior towards EBP.

This study has shown how the variables of awareness, knowledge, attitudes, and behavior towards EBP affect how physiotherapists perceive the concept and apply it in their practice. Increasing the EBP awareness will directly impact the positive outcomes on quality of both clinical and research applications of physiotherapy.

## Conclusions

This study has shown that though a sizeable number of physiotherapists in the UAE claim they are aware about EBP, their knowledge is limited to a few key terms. Attention is needed to improve EBP knowledge and practice by increasing the awareness of the concept, encouraging positive attitudes towards it, building better knowledge on the subject, and leading to positive behavior. These facilitating factors will help in controlling the barriers of lack of resources, lack of time, lack of library resources, knowledge, and skills, and lack of support.

## Supporting information

**S1 Appendix. Questionnaire.**
(PDF)

## Acknowledgments

The authors would like to express sincere gratitude for all committed participants in this study.

## Author Contributions

**Conceptualization:** Hamda AlKetbi, Fatma Hegazy, Tamer Shousha.

**Data curation:** Hamda AlKetbi, Tamer Shousha.

**Formal analysis:** Hamda AlKetbi, Tamer Shousha.

**Investigation:** Tamer Shousha.

**Methodology:** Hamda AlKetbi, Tamer Shousha.

**Project administration:** Hamda AlKetbi, Fatma Hegazy, Arwa Alnaqbi, Tamer Shousha.

**Software:** Arwa Alnaqbi.

**Supervision:** Fatma Hegazy, Tamer Shousha.

**Validation:** Hamda AlKetbi, Tamer Shousha.

**Visualization:** Hamda AlKetbi, Tamer Shousha.

**Writing – original draft:** Hamda AlKetbi, Arwa Alnaqbi, Tamer Shousha.

**Writing – review & editing:** Hamda AlKetbi, Tamer Shousha.

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
