## [Decision Letter · Decision Letter 0]

17 May 2021

PONE-D-21-13713

Evidence-based practice by physiotherapists in UAE: Investigating behavior, attitudes, awareness, knowledge and barriers

PLOS ONE

Dear Dr. Shousha,

Thank you for submitting your manuscript to PLOS ONE. After careful consideration, we feel that it has merit but does not fully meet PLOS ONE’s publication criteria as it currently stands. Therefore, we invite you to submit a revised version of the manuscript that addresses the points raised during the review process.

We look forward to receiving your revised manuscript.

Kind regards,

Walid Kamal Abdelbasset, Ph.D.

Academic Editor

PLOS ONE

Journal Requirements:

2. Please provide additional details regarding survey participant consent. In the ethics statement in the Methods and online submission information, please ensure that you have specified (1) whether consent was informed and (2) what type you obtained (for instance, written or verbal, and if verbal, how it was documented and witnessed).

3. Please upload a copy of Supporting Information Figure 1 and 2 and Supporting Table 1, 2 and 3 which you refer to in your text on page 21.

4. Please update your submission to use the PLOS LaTeX template. The template and more information on our requirements for LaTeX submissions can be found at http://journals.plos.org/plosone/s/latex.

Reviewers' comments:

Reviewer's Responses to Questions

**Comments to the Author**

1. Is the manuscript technically sound, and do the data support the conclusions?

Reviewer #1: Yes

Reviewer #2: Partly

2. Has the statistical analysis been performed appropriately and rigorously? 

Reviewer #1: Yes

Reviewer #2: No

3. Have the authors made all data underlying the findings in their manuscript fully available?

Reviewer #1: Yes

Reviewer #2: No

4. Is the manuscript presented in an intelligible fashion and written in standard English?

Reviewer #1: Yes

Reviewer #2: No

5. Review Comments to the Author

Reviewer #1: Thank you for giving me the opportunity to review this article titled “Evidence-based practice by physiotherapists in UAE: Investigating behavior, attitudes, awareness, knowledge and barriers”.

Abstract:

1. Define the acronym, when it is used in first time.

2. In the results, mention the percentage of each items answered by the participants, rather using superficial, lower ratings etc…

3. Include a clear and precise conclusion of this analysis.

Manuscript:

4. The reference number -9 deals with undergraduate students, how come it can be generalized to the whole physiotherapy population?

5. The EBP process used in this article was outdated one (1997).

6. Define the acronym, when it is used in first time (UAE, ERWCPT, WCPT etc…).

7. Attach the reference of newly validated and developed questionnaire.

8. Include the reliability and validity of newly developed questionnaire.

9. In discussion, add and identify the causes of lack of awareness and knowledge of EBP with latest references.

10. Add the clinical significance of this article.

Reviewer #2: I was happy to review this manuscript and appreciate the effort you put in this work. Meanwhile, I have some comments to consider as follow:

- The abbreviation (DHA) in the 1st affiliation is not clear.

- The 2nd affiliation is clear and not complete.

- The sentence in lines 45 & 46 is not clear.

- The sentence in lines 60 & 61 is not clear.

- The section UAE Physiotherapists in the Methods section is suitable or fits the Introduction more than the Methods section.

- Methods section is missing the statistical part.

- Line 202: I believe the test was used is the internal consistency (Cronbach alpha) not the inter-rater correlation, Please double check.

- Lines 281-288: This table is not as important as discussing the variables of the study (awareness, attitude, behavior, knowledge, and barriers) among seniors vs juniors and master holders vs baccalaureate and so on.

6. PLOS authors have the option to publish the peer review history of their article (what does this mean?). If published, this will include your full peer review and any attached files.

Reviewer #1: **Yes: **Gopal Nambi

Reviewer #2: No

---

## [Author Response · Author response to Decision Letter 0]

25 May 2021

Responds to the reviewer’s comments:

Reviewer # 1

Response to comment: Abstract:

1. Define the acronym when it is used in first time:

Response: The full term for the UAE has been added as “United Arab Emirates” in Page 2 line 41

2. In the results, mention the percentage of each items answered by the participants, rather using superficial, lower ratings.

Response: Main results were added “The attitude towards EBP was found to be significantly related to the knowledge of EBP (r = 0.208) and the perception of barriers to it (r = 0.156). The EBP behavior was found positively related to its knowledge (r=0.134) and the perception of barriers (r= 0.216).” Page 2 line 48-50.

Kindly note that it was not practical to add all results in the abstract based on the vast number of items in the questionnaire, thus we added the main to follow the reviewers’ recommendation.

3. Include a clear and precise conclusion of this analysis.

Response: Conclusion added “This study concluded that although physiotherapists in the UAE claim awareness about EBP, their knowledge is limited to a few key terms whereas, attention is needed to improve EBP knowledge and practice.” Page 2-3 line 56-58

Manuscript:

4. The reference number -9 deals with undergraduate students, how come it can be generalized to the whole physiotherapy population?

Response: The intention was not to generalize but the reference was used for 2 reasons:

A. The questionnaire is based partly on the perception EBP starting from the educational stage. 

B. This reference was used to support the need of this study in the UAE as stated, “The researcher has chosen the UAE for the setting of this study as despite being one of the more developed nations with a high per capita income, its healthcare is perceived to be weak in its implementation of EBP” Page 4 lines 83-86

5. The EBP process used in this article was outdated one (1997).

Response: The process adopted is considered one of the basic foundations still used for the EBP process and it has been cited in recent literature with example given below:

• Evidence-Based Medicine: How to Practice and Teach EBM, 2nd Edition: By David L. Sackett, Sharon E. Straus, W. Scott Richardson, William Rosenberg, and R. Brian Haynes, Churchill Livingstone, 2000

Roger LuckmannFirst Published May 1, 2001 Book Review

https://doi.org/10.1177/088506660101600307

• Practical Implementation in Social Work Practice: A Guide to Engaging in Evidence based Practice; Jennifer L. Bellamy and Danielle E. Parish, Oxford, 2020.

• EBM 

 Evidence to practice & practice to evidence; J Eval Clin Pract. 2008 Oct; 14(5): 656–659.

 doi: 10.1111/j.1365-2753.2008.01043.x

6. Define the acronym when it is used in first time (UAE, ERWCPT, WCPT etc…).

Response: Added full terms as follows:

“United Arab Emirates” in Page 2 line 41

“Europe Region World Physiotherapy” Page 5 line 104-105

“World Confederation of Physiotherapy” Page 7 line 145

“Gulf Cooperation Council” Page 7 line 150 - 151

7. Attach the reference of newly validated and developed questionnaire.

Response: Only one question was added to the questionnaire regarding the years of experience which we believed was crucial to the EBP perception. Rephrasing done to clarify “This questionnaire was further refined by adding a question regarding the years of experience [1] and validated through a pilot study with 10 physiotherapists which helped to rephrase and review some of the items” as well as adding the reference [1]. Page 8 line 170-172

The item “15 closed-ended question” was edited Page 9 line 192. 

The item” and years of experience” was added in the demographic data Page 9 line 194.

Assessment of validity added “The construct validity was assessed using Spearman rank coefficients between the modified questionnaire and the original used by AlShehri et al., 2017. The values were interpreted as follows: excellent relationship, >0.91; good, 0.90–0.71; moderate, 0.70–0.51; fair, 0.50–0.31; and little or none, < 0.30 The level of significance in all tests was set at p < 0.01” Page 10 line 220-222.

Appendix (A) updated

8. Include the reliability and validity of newly developed questionnaire: 

Response: “reliability is high as denoted by the Cronbach’s alpha value of 0.867. Therefore, the questionnaire satisfied the reliability criteria.” [already stated in text Page 11 Line 233]

Added “The validity of the questionnaire was reported as (0.92) revealing an excellent positive correlation with the original questionnaire. “Page 11 Line 235-236

9. In discussion, add and identify the causes of lack of awareness and knowledge of EBP with latest references.

Response: The discussion section has been augmented and restructured, Page 18-19 line 326-366

10. Add the clinical significance of this article.

Response: Added “This study has shown how the variables of awareness, knowledge, attitudes, and behavior towards EBP affect how physiotherapists perceive the concept and apply it in their practice. Increasing the EBP awareness will directly impact the positive outcomes on quality of both clinical and research applications of physiotherapy.” Page 20 line 383-386

Removal of “This study has several implications for the research and the practitioner communities so that the barriers to the application of EBP can be addressed while the positive relationships identified in the study can be bolstered. “Page 20 line 386-389

Reviewer # 2:

Response to comment:

1. The abbreviation (DHA) in the 1st affiliation is not clear.

Response: The abbreviation of Dubai Health authority (DHA) is listed in the PLoS one menu when entering the affiliation

2. The 2nd affiliation is clear and not complete.

Response: Affiliation is listed in the PLoS one menu when entering the affiliation, however affiliation 3 was completed.

3. The sentence in lines 45 & 46 is not clear.

Response: Rephrased “the results reveal that the awareness of EBP is superficial with more scientific and research-related terms finding lower ratings among the respondents results show that the awareness of EBP is limited to a few terms including EBP, systematic literature review, and random trials while other terms associated with scientific studies are not known well.” Page 2 line 45-47.

4. The sentence in lines 60 & 61 is not clear.

Response: Rephrased “Involved instead of applying” Page 3 line 67

5. The section UAE Physiotherapists in the Methods section is suitable or fits the Introduction more than the Methods section.

Response: The UAE physiotherapists were placed in the methods section as they are the target population of this study, and we humbly believe it is acceptable based on the nature of the current study.

6. Methods section is missing the statistical part.

Response: Added” The reliability and validity of the questionnaire were assessed using the Cronbach alpha and the Spearman’s correlation coefficient respectively with a P value set at P<0.01. The results of the demographic variables’ influence on the awareness, knowledge, attitude, and behavior towards EBP were analyzed using MANOVA where the means and standard deviations of the answered questions of the subscales were calculated and finally the relationship between different variables was assessed using the Pearson’s correlation coefficient.” Page 8-9 line 186-191

7. Line 202: I believe the test was used is the internal consistency (Cronbach alpha) not the inter-rater correlation, Please double check.

Response: Typo error “removed the inter-rater correlation, the Cronbach alpha was actually stated” Page 10 line 217

8. Lines 281-288: This table is not as important as discussing the variables of the study (awareness, attitude, behavior, knowledge, and barriers) among seniors vs juniors and master holders vs baccalaureate and so on.

Response: Added” Regarding education, the qualifications of the respondents were significantly related to the dependent variables though only knowledge of EBP differed significantly among the dependent variables. The effect size for this relationship was again low which explained 10.7% of variance in scores. The PhD holding physiotherapists had a better knowledge of EBP (Mean 66.5) than the others with each level of education contributing to higher knowledge among the physiotherapists. For instance, the Baccalaureate students revealed a Mean value of 46.49, Master’s 50.75, and DPT 53.88. This is an encouraging trend as it shows that the educational qualifications are contributing to better knowledge of EBP among the physiotherapists.

In addition, results revealed that gender did not influence the opinions of the physiotherapists towards EBP for their awareness, knowledge, attitudes or behavior. However, gender was found to have an impact on the perception of barriers in the path of implementation of EBP. Though this relationship was found to have a small effect size, the women (42.19±12.41) perceived more barriers for EBP compared to men (38.2±13.04).

Similarly, the perception of Barriers to EBP were significantly related to the age of the physiotherapists. Further exploration revealed that the differences lay between the youngest group of 20-25 years old physiotherapists and the 31-35 years and 36-40 years old physiotherapists. The effect size was found to be small with the 20-25 years old physiotherapists (30±14.39) having significantly lower perception of barriers compared to the 31-35 (41.66±13.19) and 36-40 years old physiotherapists (41.90±12.31). This result clearly shows that with younger age, the physiotherapists are encountering more barriers in the implementation of EBP.” Page 16-17 line 300-319

---

## [Decision Letter · Decision Letter 1]

31 May 2021

Evidence-based practice by physiotherapists in UAE: Investigating behavior, attitudes, awareness, knowledge and barriers

PONE-D-21-13713R1

Dear Dr. Shousha,

We’re pleased to inform you that your manuscript has been judged scientifically suitable for publication and will be formally accepted for publication once it meets all outstanding technical requirements.

Kind regards,

Walid Kamal Abdelbasset, Ph.D.

Academic Editor

PLOS ONE

Additional Editor Comments (optional):

Reviewers' comments:

Reviewer's Responses to Questions

**Comments to the Author**

1. If the authors have adequately addressed your comments raised in a previous round of review and you feel that this manuscript is now acceptable for publication, you may indicate that here to bypass the “Comments to the Author” section, enter your conflict of interest statement in the “Confidential to Editor” section, and submit your "Accept" recommendation.

Reviewer #1: All comments have been addressed

Reviewer #2: All comments have been addressed

2. Is the manuscript technically sound, and do the data support the conclusions?

Reviewer #1: Yes

Reviewer #2: Yes

3. Has the statistical analysis been performed appropriately and rigorously? 

Reviewer #1: Yes

Reviewer #2: Yes

4. Have the authors made all data underlying the findings in their manuscript fully available?

Reviewer #1: Yes

Reviewer #2: Yes

5. Is the manuscript presented in an intelligible fashion and written in standard English?

Reviewer #1: Yes

Reviewer #2: Yes

6. Review Comments to the Author

Reviewer #1: Ireally appreciate the efforts of all the authors.

The authors have addressed all the comments raised by me and it is now suitable for publication.

Reviewer #2: Thank you very much for addressing all comments. I appreciate discussing the distribution of variables among demographic variables. I wish you all the best.

7. PLOS authors have the option to publish the peer review history of their article (what does this mean?). If published, this will include your full peer review and any attached files.

Reviewer #1: **Yes: **Gopal Nambi

Reviewer #2: No

---

## [Editor Report · Acceptance letter]

8 Jun 2021

PONE-D-21-13713R1 

Evidence-based practice by physiotherapists in UAE: Investigating behavior, attitudes, awareness, knowledge and barriers 

Dear Dr. Shousha:

I'm pleased to inform you that your manuscript has been deemed suitable for publication in PLOS ONE. Congratulations! Your manuscript is now with our production department. 

Kind regards, 

on behalf of

Dr. Walid Kamal Abdelbasset 

Academic Editor

PLOS ONE